# Investigating Parents' Attitudes towards the Use of Child Restraint Systems by Comparing Non-Users and User Parents

**Thanapong Champahom** [1], **Sajjakaj Jomnonkwao** [2], **Woraanong Thotongkam** [1], **Pornsiri Jongkol** [3], **Porntip Rodpon** [1] **and Vatanavongs Ratanavaraha** [2,*]

1   Department of Management, Faculty of Business Administration, Rajamangala University of Technology Isan, Nakhon Ratchasima 30000, Thailand
2   School of Transportation Engineering, Institute of Engineering, Suranaree University of Technology, Nakhon Ratchasima 30000, Thailand
3   School of Industrial Engineering, Institute of Engineering, Suranaree University of Technology, Nakhon Ratchasima 30000, Thailand
*   Correspondence: vatanavongs@g.sut.ac.th

**Abstract:** In developing countries, there are no laws to enforce child safety seat use, so there is still a very low rate of use. This study aimed to understand parents' attitudes toward CRS use based on the health belief model (HBM) theory. To find realistic policies encouraging the use of CRSs, the model was split into two sub-models: a group of parents using a CRS (CRS user) and a group of parents not using a CRS (CRS non-user). Using confirmatory factor analysis (CFA), structural equation modeling (SEM), and measurement invariance (MI) to test the differences between the two parent groups, the CFA results indicated that there were six constructs based on the HBM. According to the individual models of SEM, in the CRS non-user model, no significant latent construct was found to affect the use of CRSs, whereas in the CRS user model, the perceived severity and the cues to action were significant for using a CRS ($p < 0.05$). The MI results indicated that the attitudes of the two parent groups were different. The recommendations for policies obtained from the study results include promotion aimed toward increasing safety awareness, public relations regarding CRS usefulness, and pricing strategies from the government sector.

**Keywords:** child safety seat; booster seats; multigroup analysis; developing country; public policy; health belief model

## 1. Introduction

Road traffic crashes are responsible for a significant number of deaths, injuries, and disabilities worldwide. Among the most vulnerable road users, children in developing countries are at a higher risk of suffering due to the insufficiency of public education and awareness (i.e., on road safety measures, accident reporting systems, public policy, law, and enforcement of road safety regulations) [1].

In Thailand, road crashes are the second leading cause of child deaths after accidental drowning and submersion. According to the Department of Disaster Prevention and Mitigation (Ministry of the Interior in Thailand), statistics show that 20,342 (approximately 7.81%) of the children below 15 years of age were seriously injured passengers in 2015–2020 (on average, 3390 per year). Notably, regarding the involved vehicle types, 1860 cases (9.14%) of severely injured children involved pickup trucks, and 856 cases (4.22%) involved passenger cars; 738 victims (3.63%) were involved in fatal road crashes [2].

A child restraint system (CRS) includes a child safety seat and booster seat designed specifically to protect children from severe or fatal injury in vehicle crashes [3]. A study in Greece found that children aged 0–11 years sitting in front seats and not using child safety seats had a greater risk (up to 5 times) of injury than those in the rear seat using safety seats [4]. According to the World Health Organization, the utilization of safety seats

can significantly decrease the incidence of infant mortality by 60%. Furthermore, it is noted that laws concerning the implementation of child restraint systems in various nations are in compliance with established best practices. It has been observed that a majority of high-income countries, at 85%, adhere to these best practices, whereas the minority of middle-income countries, at 15%, also implement such laws [5].

According to a safety seat use survey of a sample group in China, the factors affecting child safety seat use were the perception of usefulness and public accessibility to products. This suggests that the involved agencies need to have public relations and added information about the benefits of child safety seats added into the driving school curriculum and driving license permissions to let parents know about the importance of child safety seats in cars [6–8].

According to an evaluation conducted by the World Health Organization in Thailand, the nation's laws pertaining to road safety are in compliance with international standards, including regulations for seat belt usage, speed limits, and helmet usage. However, it was noted that there is currently no legal enforcement for the use of child safety seats in vehicles [5]. In addition, people are still unaware of the importance of using CRSs in cars. Previously, Thailand has established a network for children's safety. It consists of government agencies, private sectors, and educational institutions that conduct studies to find ways to help support equipment application, including only seat belts and helmets, in addition to measures and campaigns for the widespread use of such protective equipment. The mentioned measures aim to reduce the severity of road crashes; however, there have been limited studies on CRSs.

The parents are the ones who decide whether or not to use a CRS. Therefore, to focus on measuring the motivation for using a child safety seat, theories should focus on behavior, attitudes, motivation, and perceptions [9,10]. In previous studies, there have been many theories on reducing road accident severity. However, the most widely used and confirmed theory that can apply to the intention of safety equipment use is the health belief model theory, such as the study of the motivation for using seatbelts [11] and helmets [10,12].

Previous studies have focused on the attempt to find ways to increase the use of CRSs in cars [3,13–15], which have included car seats, booting restraints, and seat belts (Table 1). For the overall picture, most researchers have studied parents' social and economic factors, [8,16] as well as their attitudes [1,17,18] toward the use of CRSs. However, only one [19] has studied the analytical dimension based on a psychological model. Its outstanding fundamental analysis includes the presence of clear hypotheses or research questions [10]. In addition, for parent groups, researchers should include both CRS user and CRS non-user groups, as there have been no studies previously comparing models of these two groups. Thus, this study seeks to fulfill this knowledge component.

This study aimed to compare whether or not there is a difference between the existing CRS user and CRS non-user groups in addition to comprehending parents' attitudes toward the use of safety child seats. The results can potentially be used to develop guidelines for promoting and publicizing the use of child safety seats in cars in Thailand for public awareness of the importance of preventing children from experiencing road accidents.

**Table 1.** Comparison of studies relating to child restraint use.

| Literature | Data Collection Area | Child Restraint Type | | | Aims of the Study | Theory | Method |
|---|---|---|---|---|---|---|---|
| | | CSS | BS | SB | | | |
| Bryant-Stephens et al. [13] | Pennsylvania, USA | | ✓ | | To evaluate the effectiveness of a theoretically grounded, community-delivered marketing campaign to promote belt-positioning booster seats | | Generalized linear model |

**Table 1.** *Cont.*

| Literature | Data Collection Area | Child Restraint Type | | | Aims of the Study | Theory | Method |
|---|---|---|---|---|---|---|---|
| | | CSS | BS | SB | | | |
| Charlton et al. [16] | New South Wales and Victoria, Australia | | ✓ | ✓ | To design to gather more knowledge about restraint usage rates, patterns of restraint usage, and 'appropriateness' of restraint use by children of the 'booster seat age', as well as the attitudes of parents of children in the booster seat age group towards restraint-wearing behavior | | T-tests and chi-squares |
| Chaudhry et al. [1] | Pakistan | ✓ | | | To identify problems that influence non-usage among knowledgeable parents, as well as to explore the need and interventions for conducting effective awareness campaigns or educational programs for parents in the country | | Spearman correlations and logistic regression analysis |
| Chen et al. [17] | Shantou, China | ✓ | ✓ | ✓ | To describe child restraint use and examine the driver's knowledge of and attitude toward the use of child restraints | | Multivariable regression |
| Techakamonsuk [19] | Thailand | ✓ | | | To identify the factors supporting effective child safety seat use and to develop effective policies | PRECEDE-PROCEED | Chi-square test |
| Elhalik et al. [18] | Dubai, UAE | ✓ | | | To evaluate maternal awareness and perception on child car safety seat usage | | Chi-square and Fisher's test |
| Jarahi et al. [14] | Mashad, Iran | ✓ | | | To assess parental willingness to pay for child car safety seats in Mashad, Iran | | Logistic regression model |
| Liu et al. [7] | Shantou, China | ✓ | | | To investigate the knowledge, attitudes, and intended behaviors about use of child safety seats among parents of newborns and explore expectant mothers' views and decisions regarding child safety seats use | | A qualitative and quantitative (logistics regression) approach |
| Liu et al. [8] | Shantou, China | ✓ | | | To determine the effect of intervention on the improvement of knowledge of child passenger safety and use of CSS among parents with newborns. We hypothesized that parents assigned into the intervention group who received education and free CSS would have greater increases in child passenger safety knowledge and use of CSS, as compared to parents in the education only or control groups | | Chi-square tests and logistic regression |

**Table 1.** *Cont.*

| Literature | Data Collection Area | Child Restraint Type | | | Aims of the Study | Theory | Method |
|---|---|---|---|---|---|---|---|
| | | CSS | BS | SB | | | |
| Tan et al. [15] | Singapore | ✓ | | | To understand parental knowledge, beliefs, and barriers regarding the use of child car restraints (CCRs) | | a qualitative study |
| Ramli et al. [3] * | Malaysia | ✓ | | | (1) To determine the prevalence of the use of CRSs in Malaysia, (2) to evaluate injuries related to unrestrained children, and (3) to show the nation's preparation towards implementation of a child restraint law | | |
| This study | Thailand | ✓ | ✓ | | To understand the attitude of parents toward child safety seat use by comparing parent groups that use and do not use CSS | Health belief model | Measurement of invariance confirmatory factor analysis |

Note: * Review article; CSS denotes child safety seat; BS denotes booster seat; and SB denotes seatbelt.

## 2. Health Belief Model

The health belief model (HBM) is a concept that has been generally used to explain the influencing factors on individual health behaviors that individuals may find and adopt regarding a certain preventive health strategy with an early consensus (e.g., checkups or rehabilitation).

Rosenstock [20] classified the HBM into four components: perceived susceptibility to disease, perceived severity of disease, perceived benefits of preventive action, and perceived barriers to preventive action. Because previous HBMs could only predict preventive disease behavior, Maiman and Becker [21] extended these factors and added more details as described below. In this research, "disease" means a reduction in childhood mortality in road crashes. The HBM has the following factors [12]:

(1). Perceived benefits of preventive action focuses on how people look for ways to stay healthy, prevent disease, or recover from it. Such practices lead to useful approaches to sensitivity prevention, and the decision to manifest such behaviors is based on weighing the benefits and drawbacks of the behavior. This refers to the perception of the benefits of using a CRS. For example, if using a child safety seat, it is not necessary to carry the child while in the car, and if using a child safety seat, it will be comfortable to take care of children in the car [22].

(2). Health motivation refers to the emotional state that is stimulated by health-related matters, such as levels of attention, care, attitudes, and health values. For the use of a CRS, this would refer to giving weight to the road crashes that affect children's safety (for example, getting in a road accident being the worst) or to the view that children's health is the most important, and giving importance to the safety of your child/children when driving [7,18].

(3). Perceived susceptibility refers to individuals' direct beliefs regarding their own behavior. They believe or forecast that their disease avoidance measures are related to their chance of developing a disease or having some level of health difficulty. The perceived susceptibility to a disease is widely regarded as the most important factor influencing people's attitudes toward good health. Regarding use of a CRS, with parents' confidence in driving, a crash would not occur; as a result, there is no need to use a CRS. The question items are as follows: "I think a child safety seat is not needed when driving to nearby places"; "I have many years of driving experience";

"I can avoid crashes"; or "I think that a child safety seat is not quite important for experienced drivers" [9,23].

(4). Perceived severity of disease refers to people's beliefs about their ability to evaluate the severity of diseases or health problems. Furthermore, the causes of disabilities, death, difficulties, time-consuming cures, complicated diseases, or effects on social life are factors in determining perceived severity. However, the perceived susceptibility to disease correlates with the perceived severity of disease, allowing people to recognize and avoid the perceived threat of disease. From the CRS users' viewpoint, it means that the impact of crashes will affect children. The question items are as follows: "In case of an accident where a child/children is/are not in child safety seats, it will affect the feelings of people I know, such as parents and elder relatives"; "In case of an accident where a child/children is/are not in a car safety seat, it may lead to deaths or disabilities as well as long-term treatment" [24].

(5). Perceived barriers to preventive action refers to individuals' beliefs in the possible obstacles that may influence their behavior. These beliefs are linked to negative health outcomes, such as healthcare expense and disease. As a result, they believe that the presence of numerous problems and obstacles will cause difficulty regarding their behavioral changes. Their perception of barriers to using a CRS refers to the difficulty of use and the cost of a CRS. The question items are as follows: "I think child safety seats are more expensive than their values or benefits"; "I think a good quality child safety seat is too costly for me to afford"; and "Installing a safety seat in a car is a hassle for me" [22,25].

(6). Cue to action refers to the events that encourage an individual's required behavior. These cues can be internal and external factors. This is the stimulus needed to trigger the decision-making process to accept a recommended health action. To reflect the aims of this study (development of guidelines for promotion and publicity), the questions focus on only the external factors; therefore, the questionnaires are based on promotion [14]. For example, from a hospital perspective, "Hospitals should provide child car seats for sale/rent/loan to the mother after giving birth," and from the parents' and surrounding peoples' perspective, "A close friend thinks I should use a CRS when travelling" [15].

(7). Modifying factors refer to those factors that have an indirect impact on health behavior through recognition and practice (demographic variables such as age, income, occupation, and educational level, as well as sociopsychological factors, such as health motivation, might influence an individual's decision to use a CSS). However, the researchers did not analyze the mentioned factors due to their existence in multiple studies, such as that by Techakamonsuk [19].

## 3. Materials and Methods

### 3.1. Questionnaire Design

The questionnaire consists of two parts. The first part is about general information, transportation behavior data, and traveling behavior data [26]. The second part consists of question items regarding attitudes toward CRS use, which had been studied in many research studies, and was designed by reviewing the meanings of the factors of the HBM based on Section 2 (the questions shown in Table 2). To measure the observed variables for the HBM, the scales were measured using the attitudes and beliefs of the respondents, or by allowing the respondents to provide their overall points of view for each question [27]. Specifically, the scale used in this study was based on a five-point Likert-type scale (1 = completely disagree and 5 = completely agree) [23,28]. In this study, the evaluation of concepts is focused on determining their internal consistency reliability using Cronbach's alpha. This statistic gauges the consistency of responses across the elements within a measure, and is used as a measure of internal consistency reliability. If the internal consistency is found to be low, it may indicate that the items being evaluated are too diverse in content,

making the total score less effective as a unit of analysis for the measure [29]. A Cronbach's alpha > 0.6 is still within the acceptable range [30].

**Table 2.** Questionnaire description.

| Variable | Description | CRS Non-User Mean | S.D | CRS User Mean | S.D | *t*-Test for Equality of Means t | df | *p*-Value |
|---|---|---|---|---|---|---|---|---|
| | **Perceived Benefits** | | | | | | | |
| PB1 | I think, if using a child restraint system, it is not necessary to carry the child while in the car. | 4.42 | 0.64 | 4.51 | 0.61 | −2.05 | 779.28 | 0.04 |
| PB2 | I think, if using a child restraint system, it will be comfortable to take care of children in the car. | 4.51 | 0.60 | 4.49 | 0.64 | 0.45 | 744.39 | 0.66 |
| PB3 | I think using a child restraint system can reduce crash injury severity. | 4.50 | 0.64 | 4.54 | 0.62 | −1.03 | 773.05 | 0.30 |
| | **Health motivation** | | | | | | | |
| HM1 | I think being involved in a road crash is the worst. | 3.90 | 0.78 | 3.92 | 0.75 | −0.22 | 775.12 | 0.82 |
| HM2 | I think the health of my child/children is the most important. | 3.93 | 0.75 | 3.89 | 0.77 | 0.70 | 761.55 | 0.48 |
| HM3 | I give great importance to my child/children's safety when driving. | 3.82 | 0.82 | 3.83 | 0.81 | −0.18 | 771.60 | 0.86 |
| | **Cue to Action** | | | | | | | |
| CA1 | I think that hospitals should provide child car seats for sale/rent/loan to the mother after giving birth. | 4.46 | 0.67 | 4.47 | 0.70 | −0.26 | 756.90 | 0.80 |
| CA2 | My close friend thinks I should use a child restraint system when I travel. | 3.05 | 0.70 | 3.06 | 0.73 | −0.18 | 754.82 | 0.85 |
| CA3 | I think that the government should promote the use of child restraint systems by supporting the purchases. | 4.76 | 0.50 | 4.78 | 0.47 | −0.49 | 779.88 | 0.62 |
| | **Perceived Susceptibility** | | | | | | | |
| PSU1 | I think a child restraint system is not needed when driving to nearby places. | 2.83 | 0.70 | 2.74 | 0.68 | 1.69 | 773.22 | 0.09 |
| PSU2 | I have years of driving experience; I can avoid crashes. | 2.87 | 0.74 | 2.82 | 0.73 | 0.89 | 769.82 | 0.38 |
| PSU3 | I think a child restraint system is not quite important for experienced drivers. | 2.73 | 0.71 | 2.70 | 0.75 | 0.47 | 751.31 | 0.64 |
| | **Perceived Severity** | | | | | | | |
| PSE1 | In case of a crash where a child/children is/are not in a child restraint system, it may affect their education. | 4.00 | 0.74 | 3.94 | 0.74 | 1.25 | 769.33 | 0.21 |
| PSE2 | In case of a crash where a child/children is/are not in a safety seat, it will affect the feelings of people I know, such as parents, elder relatives, etc. | 3.88 | 0.78 | 3.88 | 0.76 | −0.11 | 775.46 | 0.91 |
| PSE3 | In case of a crash where a child/children is/are not in a child restraint system, it may affect their future. | 3.79 | 0.74 | 3.76 | 0.70 | 0.63 | 779.34 | 0.53 |
| | **Perceived Barriers** | | | | | | | |
| PBA1 | I think child restraint systems are more expensive than the value or benefits they offer. | 4.44 | 0.71 | 4.33 | 0.73 | 2.15 | 760.34 | 0.03 |
| PBA2 | Installing safety seats in a car is a hassle for me. | 3.86 | 1.08 | 3.90 | 0.98 | −0.52 | 798.00 | 0.60 |
| PBA3 | I think a child restraint system of good quality is too costly for me to afford. | 4.42 | 0.68 | 4.38 | 0.72 | 0.82 | 751.55 | 0.41 |

### 3.2. Data Collection

Sample size: The rule of calculating sample size in structural equation modeling (SEM) is to obtain a large sample size relevant to the number of factors in the analysis. Using the maximum likelihood estimation, in the simple model, there should be at least 200 samples or a sample size at least equal to 15 times the indicators [31]. In this study, there were 19 observed variables (18 independent variables and 1 dependent variable, for an individual model), so the sample size should be at least $19 \times 15 = 258$ samples [32,33].

Participants: The researchers conducted this study by collecting data from areas throughout the country. The representatives were selected from large cities in each region in Thailand, because they tend to have a higher number of CRS users than secondary provinces. The study areas for data collection included five provinces, namely, Bangkok, Chiang Mai, Chon Buri, Nakhon Ratchasima, and Songkhla (representatives of the central region, northern region, eastern region, northeastern region, and southern region, respectively). Most of the collected data were from school fronts, shopping malls, and hospitals in the city districts of the province. Initially, data from 815 participants were collected; however, 15 observations were excluded due to incomplete data (e.g., duplicate, invalid, and missing data).

The participant characteristics presented in Table 3 divided parents into two groups as follows: (1) CRS non-users, which refers to parents using child safety belts and carrying small children (n = 440), and (2) CRS users, which refers to those who use a CRS whenever they travel (n = 360). The sample characteristics of the two groups are relatively similar. For example, most of the participants have bachelor's degree; their income is mostly 20–30 k per month; most of the commuters are parents; most of them are private company employees; and most of the cars are four-door pickups or cars. Chonburi and Nakhon Ratchasima had the highest proportion of CRS users (9.3%). Regarding the frequency of using CRS, 33.25% of the user group answered "always", whereas all CRS non-users answered "never".

**Table 3.** Sample characteristics.

| Characteristic | | CRS Non-User | | CRS User | |
| --- | --- | --- | --- | --- | --- |
| | | Count | % | Count | % |
| Gender | Male | 254 | 31.80% | 171 | 21.40% |
| | Female | 186 | 23.30% | 189 | 23.60% |
| Average age (years) mean ± S.D. | | 36.18 ± 9.92 | | 35.88 ± 9.22 | |
| Average children age (years) mean ± S.D. | | 2.96 ± 1.80 | | 2.66 ± 2.02 | |
| Child relationship | Parent | 300 | 37.50% | 228 | 28.50% |
| | Relative | 140 | 17.50% | 132 | 16.50% |
| Education | Primary school | 41 | 5.10% | 30 | 3.80% |
| | Junior high school | 65 | 8.10% | 56 | 7.00% |
| | High school | 65 | 8.10% | 43 | 5.40% |
| | High vocational | 33 | 4.10% | 22 | 2.80% |
| | Bachelor's degree | 211 | 26.40% | 196 | 24.50% |
| | Master's degree and higher | 27 | 3.10% | 13 | 1.70% |
| Occupation | Government officer | 10 | 1.30% | 11 | 1.40% |
| | Private sector | 126 | 15.80% | 113 | 14.10% |
| | Private business | 167 | 20.90% | 121 | 15.10% |
| | Agriculturist | 59 | 7.40% | 57 | 7.10% |
| | Student | 18 | 2.30% | 19 | 2.40% |
| | General employee | 60 | 7.50% | 39 | 4.90% |

**Table 3.** *Cont.*

| Characteristic | | CRS Non-User | | CRS User | |
|---|---|---|---|---|---|
| | | **Count** | **%** | **Count** | **%** |
| Salary (THB per month) | 10,001–20,000 | 11 | 1.40% | 10 | 1.30% |
| | 20,001–30,000 | 114 | 14.30% | 94 | 11.80% |
| | 30,001–40,000 | 71 | 8.90% | 77 | 9.60% |
| | 40,001–50,000 | 77 | 9.60% | 60 | 7.50% |
| | 50,001–60,000 | 63 | 7.90% | 39 | 4.90% |
| | 60,6001–70,000 | 58 | 7.30% | 44 | 5.50% |
| | >70,001 | 46 | 5.80% | 36 | 4.50% |
| Urbanization | Urban | 206 | 25.80% | 103 | 12.90% |
| | Sub-urban | 105 | 13.10% | 145 | 18.10% |
| | Rural | 129 | 16.10% | 112 | 14.00% |
| Marital status | Married | 204 | 25.50% | 184 | 23.00% |
| | Others | 236 | 29.50% | 176 | 22.00% |
| Frequency of travelling with the child | Less than 1 time | 141 | 17.60% | 124 | 15.50% |
| | 1–2 times per week | 114 | 14.30% | 97 | 12.10% |
| | 3–5 times per week | 64 | 8.00% | 60 | 7.50% |
| | Every time of travelling | 121 | 15.10% | 79 | 9.90% |
| Vehicle type | Pickup | 44 | 5.50% | 32 | 4.00% |
| | Four- door pickup | 144 | 18.00% | 132 | 16.50% |
| | Car | 143 | 17.90% | 111 | 13.90% |
| | SUV | 50 | 6.30% | 33 | 4.10% |
| | Pick-up passenger vehicle | 59 | 7.40% | 52 | 6.50% |
| Province | Bangkok | 89 | 11.10% | 71 | 8.90% |
| | Chiang Mai | 88 | 11.00% | 72 | 9.00% |
| | Chonburi | 86 | 10.80% | 74 | 9.30% |
| | Nakhon Ratchasima | 86 | 10.80% | 74 | 9.30% |
| | Songkhla | 91 | 11.40% | 69 | 8.60% |
| Using CRS | 1 = Never (0%) | 440 | 55.00% | 0 | 0.00% |
| | 2 = Occasionally (approximately 25%) | 0 | 0.00% | 6 | 0.75% |
| | 3 = Sometimes (approximately 50%) | 0 | 0.00% | 28 | 3.50% |
| | 4 = Often (approximately 75%) | 0 | 0.00% | 60 | 7.50% |
| | 5 = Always (100%) | 0 | 0.00% | 266 | 33.25% |

*3.3. Methods*

3.3.1. Exploratory Factor Analysis (EFA)

The purpose of the EFA is to explain the covariance of multiple variables in terms of a few unobserved factors. The factor analysis method is appropriate for variables on interval and ratio scales, as the correlation matrix used in EFA is based on a specific statistical model. Actually, this research has already obtained various indicators and was confirmed via the HBM theory. However, two indicators, namely, promotion and law enforcement, were added, so EFA is used to regroup the indicators concerning the attitudes toward CRS use for the second time. The interpretation of the factor analysis is evaluated using factor loadings. High factor loading indicates a highly influential factor, whereas low factor loading indicates a low influential factor. Inspection of variables with strong factor loadings on a single factor is used to reveal structure or similarity across variables. The underlying constructs shared by variables with high factor loading on specific factors must then be identified [34].

### 3.3.2. Confirmatory Factor Analysis (CFA)

CFA is used to see how well the measured variables explain the constructs. The main advantage is that a concept-based theory can be analytically tested and describes how various measured items represent psychological, sociological, and business measures. When CFA results are paired with construct validity tests, researchers can improve their understanding on the quality of the measurements [29,35,36]. According to CFA model development, it initially uses the observed variables to indicate the latent variables based on the result of the EFA (initially called Model 1). Next, the model is justified using the model fit indices. If the model is not fit, the observed variables with the lowest coefficients will be removed one by one. This process is stopped when the fitted model is found. If some latent variables are measured using only two indicators (it is generally better to have at least three indicators per factor), the analysis will be prone to problems, especially in a small sample size. In addition, it will be difficult to estimate the measurement error correlation, and the factor will not be identified by itself. However, the structure of the CFA model can be identified for two indicators per latent variable condition if the errors of those two or more indicators are uncorrelated with each other or with at least one other indicator [37], which can be checked using standardized covariance residuals (correlations). The residual correlations should be less than 0.10 [29]. Many previous studies also used two indicators per factor, including those by Feng et al. [38], Allen et al. [39], Allen et al. [40], Li et al. [41], and Ni et al. [42].

### 3.3.3. Invariance Analysis of CFA and Model Specification

The purpose of the multigroup CFA model analysis is to compare models that have the same structure but different factors, such as gender, location, and culture [43]. The first section of the questionnaire was designed to examine the equivalent of two groups, also known as invariance measurement equivalence [31]. The second section evaluated measurement models using cross-validation to analyze various parameters in CFA, such as the number of constructs, indicator factor loadings, mean, and covariance. However, the statistical values of the $\chi^2$ difference (Delta $\chi^2$) and degree of freedom difference (Delta-DF) were used for evaluation [31]. The results were taken to test the significant differences between the parameter values of the two models to see whether or not an invariance existed [31,44].

This study aimed to compare the differences in the parental attitude models between CRS user parents and CRS non-user parents. The established models compared factorial invariances by starting to build the model by forcing the various parameters such as factor loading and intercept to be held in equal groups. Thus, although there were two population groups, the parameters were not different. Subsequently, the established models allowed the parameters to freely estimate the factors [29]. This method resulted in unequally valued factors for the two models [35,45]. Subsequently, the suitability of the models was compared through changing the value of the model fit indices, focusing on the values of ΔCFI, ΔTLI, $\Delta\chi^2$, and Δdf [46,47].

### 3.3.4. Structural Equation Modeling (SEM)

SEM is the typical technique used in social science research that allows for separation of the relationships for each of the sets of dependent variables, which in this case are the factors based on the HBM. SEM has two basic components: the structural model and the measurement model. The structural model is the path model, which connects the independent variables to dependent variables (i.e., variables reflecting the use of CRS). Interpretation of an individual variable's effect is achieved by examining the estimated coefficient (weight) of each variable in the variate. Readers may refer to the study by Hair et al. [48] for a more detailed explanation of SEM.

### 3.3.5. Incremental Fit Indices

The computation of the root mean squared error of approximation (RMSEA) seems to be rather straightforward, and it is offered here to highlight how statistics attempt to remedy the difficulties associated with the $\chi^2$ statistic alone where an acceptable range is $\chi^2/\mathrm{df} < 3$ and RMSEA < 0.06 [49].

Standardized root mean residual (SRMR) denotes the prediction error for each covariance term that results in a residual. SRMR was useful in evaluating the goodness-of-fit of a model compared with another model. The general rule is that an SRMR greater than 0.1 indicates a problem. The accepted SRMR is <0.08 [32,50].

The comparative fit index (CFI) is an improved version of the normed fit index that is unaffected by model complexity. It is one of the most commonly used indices. Models with CFI values greater than 0.95 are often considered well-fitting models, but those with CFI values greater than 0.90 are considered to be in the acceptable range [51].

The Tucker–Lewis index (TLI) considers and takes into account model complexity by comparing the normed $\chi^2$ values of the null model with specified models [52]. The desired TLI value is greater than 0.95, but values greater than 0.9 are within the acceptable range [53,54]. MPlus version 7 was used for data analysis [55].

## 4. Results

### 4.1. Descriptive Statistics

Table 2 presents the question items, means, standard deviations, and *t*-tests for equality of means to compare means between the two parent groups. Overall, the mean values of the question items were similar between the two groups. However, some variables were found to be significantly different between the two groups upon further analysis. Specifically, perceived benefits (PB) were found to be significantly different between the two groups, with CRS users reporting a higher average PB1 score of 4.51 compared to non-users at 4.42 ($p = 0.04$). This result aligns with the expectation that CRS users have a greater belief in the benefits of CRSs. Additionally, the perceived barriers (PBA) in the use of CRSs were found to be significantly different between the two groups, with non-users reporting a higher average PBA1 score of 4.44 compared to CRS users at 4.33 ($p < 0.03$). This result is also reasonable, as CRS non-users may perceive more barriers to the use of CRSs, which may influence their decision not to use them [25].

### 4.2. EFA Result

Table 4 presents the results of the EFA, focusing on grouping according to the HBM. The overall Cronbach's alpha value of the questionnaire is 0.602, which is relatively small, possibly due to the negative correlation of variables in the group. However, a Cronbach's alpha > 0.6 can be acceptable [30]. Furthermore, the Kaiser–Meyer–Olkin measure of sampling adequacy is equal to 0.880, additionally confirming the questionnaire reliability [56,57]. The Harman's one-factor test was applied to assess the potential for common method bias. If the total variance extracted by a single factor exceeds 50%, it may indicate a problem with common method bias. However, in this case, the total variance extracted by one factor was only 23.818%, indicating that there is no significant concern for common method bias in the data [58].

For grouping, six organized groups were found, with the Cronbach's alpha value ranging from 0.465 (Cues to Action) to 0.816 (Health Motivation). A low value of alpha could be due to a low number of questions, poor inter-relatedness between questionnaire items, or heterogeneous constructs [59]. However, some papers have indicated that a value of alpha of at least 0.45 can be acceptable [60]. The factor loadings are in the range of 0.365–0.728, showing that the lowest value is quite low. However, the values are within the acceptable range [61–63]. The selection of the number of factors can be determined by analyzing the eigenvalues and the total variance explained. It is suggested that eigenvalues greater than 1 should be considered, as any factor with an eigenvalue less than 1 explains less variation than that explained by a single variable. The results of the eigenvalue analysis

indicate that four factors had an eigenvalue of 1.22, while five factors yielded an eigenvalue of 0.986. However, this resulted in a low total variance explained of only 57.18%. Therefore, we chose to select six factors, with an eigenvalue of 0.818. It is worth noting that eigenvalues lower than 1 may be acceptable if they are close to 1 [48], with the total explained variance equaling 61.732% [64]. Rozental et al. [65] opted to select a number of factors based on the eigenvalue of 0.68. While the last factor fell below the threshold for the eigenvalue, it was retained in the final analysis due to theoretical considerations. This decision was based on the relevance of this factor to the scope of the study, specifically its alignment with the experience of CRS use in the context of the health belief model.

**Table 4.** Exploratory factor analysis result.

| Observed Variable | Cronbach's Alpha | Factor | | | | | | Variance Explained | |
|---|---|---|---|---|---|---|---|---|---|
| | | 1 | 2 | 3 | 4 | 5 | 6 | Eigenvalues | % of Variance |
| Perceived Benefits | 0.624 | | | | | | | 4.912 | 27.29 |
| PB1 | | **0.603** | −0.168 | 0.104 | −0.178 | 0.066 | 0.049 | | |
| PB2 | | **0.728** | −0.152 | −0.002 | −0.023 | −0.028 | 0.035 | | |
| PB3 | | **0.403** | −0.057 | 0.030 | −0.077 | 0.089 | 0.139 | | |
| Health Motivation | 0.816 | | | | | | | 1.711 | 9.507 |
| HM1 | | −0.395 | 0.355 | −0.126 | **0.475** | −0.116 | −0.192 | | |
| HM2 | | −0.441 | 0.305 | −0.139 | **0.460** | −0.116 | −0.222 | | |
| HM3 | | −0.468 | 0.327 | −0.051 | **0.552** | −0.164 | −0.113 | | |
| Cue to Action | 0.465 | | | | | | | 1.463 | 8.127 |
| CA1 | | 0.045 | −0.052 | 0.087 | −0.019 | 0.238 | **0.533** | | |
| CA2 | | 0.171 | −0.117 | 0.084 | −0.102 | 0.297 | **0.356** | | |
| CA3 | | 0.222 | −0.174 | 0.034 | −0.145 | 0.065 | **0.463** | | |
| Perceived Susceptibility | 0.562 | | | | | | | 1.22 | 6.78 |
| PSU1 | | 0.058 | −0.041 | **0.561** | −0.014 | 0.057 | 0.007 | | |
| PSU2 | | −0.029 | 0.058 | **0.514** | −0.062 | 0.063 | 0.096 | | |
| PSU3 | | 0.094 | 0.005 | **0.567** | −0.012 | −0.039 | 0.019 | | |
| Perceived Severity | 0.733 | | | | | | | 0.986 | 5.48 |
| PSE1 | | −0.190 | **0.627** | 0.010 | 0.175 | −0.162 | −0.023 | | |
| PSE2 | | −0.154 | **0.659** | −0.001 | 0.097 | −0.097 | −0.277 | | |
| PSE3 | | −0.132 | **0.701** | 0.058 | 0.054 | 0.091 | −0.028 | | |
| Perceived Barriers | 0.397 | | | | | | | 0.818 | 4.547 |
| PBA1 | | 0.070 | 0.013 | 0.034 | −0.077 | **0.550** | 0.049 | | |
| PBA2 | | 0.157 | −0.409 | 0.096 | −0.269 | **0.433** | 0.240 | | |
| PBA3 | | 0.011 | −0.077 | 0.011 | −0.003 | **0.537** | 0.159 | | |

Note: Harman's one-factor test for common method bias, with a percentage of variance of 23.818. Rotation method: varimax with Kaiser normalization. Cronbach's alpha all variables = 0.602. Kaiser–Meyer–Olkin measure of sampling adequacy = 0.880. Bartlett's test of sphericity = $\chi^2$ (153) = 3600.40, *p*-value < 0.000. The highest loading for each item is in bold.

### 4.3. CFA and Invariance Measurement Results

The CFA results were subsequently developed from the EFA results by choosing the best model for further SEM analysis, which was achieved by evaluating each model's fit index values (Table 5). That is, if the model failed to conform to the acceptable criteria ($\chi^2$/df < 3, CFI > 0.90, TLI > 0.90, RMSEA < 0.06, and SRMR < 0.08) or violated any of the model fit metrics, it was considered to have poor fit [29]. Model 1 included 6 latent variables with 18 observed variables, and its goodness-of-fit results, $\chi^2$/df > 3 and TLI < 0.90, indicated that it was an unsuitable model. Models 2 and 3 were subsequently built by removing PSU3 and PSE3, respectively, but both remained unsuitable as well. Finally, when PBA3 was removed to form Model 4, producing an $\chi^2$ value of 201.378 with 75 degrees of freedom, CFI = 0.949, TLI = 0.928, RMSEA = 0.046, and SRMR = 0.034, a suitable fit was observed. This model consisted of three latent factors measured using two observed variables. In addition, residual correlations of Model 4 were found in a range of 0–0.056, which also met the acceptable CFA criterion [29].

**Table 5.** Goodness of fit.

| Description | $\chi^2$ | df | $\chi^2$/df | CFI | TLI | RMSEA | SRMR | $\Delta\chi^2$ | $\Delta$df | *p*-Value |
|---|---|---|---|---|---|---|---|---|---|---|
| CFA: Entire sample | | | | | | | | | | |
| Model 1 (All observed variables) | 381.445 | 120 | 3.18 | 0.914 | 0.891 | 0.052 | 0.044 | | | |
| Model 2 (Remove PSU3) | 361.268 | 104 | 3.47 | 0.912 | 0.885 | 0.056 | 0.045 | | | |
| Model 3 (Remove PSE3) | 299.073 | 89 | 3.36 | 0.919 | 0.891 | 0.054 | 0.042 | | | |
| Model 4 (Remove PBA3) | 201.378 | 75 | 2.69 | 0.949 | 0.928 | 0.046 | 0.034 | | | |
| CFA: Measurement invariance | | | | | | | | | | |
| Model A: Factor-loading and intercept-equal groups | 304.496 | 174 | 1.75 | 0.947 | 0.937 | 0.043 | 0.049 | | | |
| Model B: Simultaneous | 263.503 | 150 | 1.76 | 0.954 | 0.936 | 0.044 | 0.038 | 40.993 | 24 | 0.017 |
| SEM: HBM individual group | | | | | | | | | | |
| CRS non-user model | 184.555 | 84 | 2.20 | 0.931 | 0.901 | 0.053 | 0.041 | | | |
| CRS user model | 166.784 | 84 | 2.99 | 0.933 | 0.904 | 0.052 | 0.042 | | | |

As presented in Table 5, regarding the multigroup CFA analysis or measurement of invariance, the goodness-of-fit of Model A (factor-loading and intercept-equal groups) generated an $\chi^2$ value of 304.496 with 174 degrees of freedom, CFI = 0.947, TLI = 0.937, RMSEA = 0.043, and SRMR = 0.049. Overall, it met the acceptable criteria [44]. However, if we independently established models for the CRS user and CRS non-user groups (Model B: Simultaneous; Table 5), the $\chi^2$ value was 263.503 with 150 degrees of freedom, CFI = 0.954, and TLI = 0.936. The model also generated RMSEA = 0.044 and SRMR = 0.038. The errors may seem worse, but the fit indices significantly improved, and the model became better when considering the CFI and TLI values. It can be initially concluded that the two population groups should not be considered in the same model. In addition, Model B also generated the value of $\Delta\chi^2$ = 40.993, with degrees of freedom of $\Delta$df = 24, which then gave a *p*-value of 0.017. This result indicates that the null hypothesis (that individual models are invariant and should not be separated into the individual model) can be rejected at a 95% confidence level. In other words, the attitudes between CRS users and non-users vary; thus, the models of CRS user and non-user parents should be separately constructed [10,53,66].

*4.4. SEM Results*

A model suitability investigation, as shown in model fit indices, found that both models were within acceptable goodness-of-fit values. Tables 6 and 7 present the SEM model results. In the individual model, the results show that 15 indicators in both parent groups were statistically significant. In the structural model, it was found that the CRS non-user model was not significant (Figure 1), whereas the CRS user model had two significant factors including cues to action (coefficient [coef.] = 0.404, *p*-value < 0.05) and perceived severity (coef. = 0.607, *p*-value < 0.05) (Figure 2). Several variables in measurement models reveal low factor loadings (e.g., PB3, CA2, and PSU2), indicating a weak causal effect between latent variables and observed indicators. This could be attributed to regional variations in attitudes towards the CRS use concept in the Thai context, leading to deviations from the theoretical framework. Nevertheless, results from SEM suggest that these factor loadings are still statistically significant. The measurement models are in line with previous findings using confirmatory factor analysis (CFA), with all indicators demonstrating statistical significance. Based on prior research, factor loadings greater than 0.20 can be deemed acceptable [39,40].

**Table 6.** Structural equation modeling results for the CRS non-user.

| Variable | CRS Non-User | | | | | |
|---|---|---|---|---|---|---|
| | Estimate | S.D. | t-Stat | *p*-Value | Lower 5% | Upper 5% |
| Measurement Model | | | | | | |
| Perceived Benefits | | | | | | |
| PB1 | 0.577 ** | 0.082 | 7.011 | <0.000 | 0.442 | 0.712 |
| PB2 | 0.487 ** | 0.074 | 6.541 | <0.000 | 0.364 | 0.609 |
| PB3 | 0.324 ** | 0.069 | 4.689 | <0.000 | 0.210 | 0.437 |
| Health Motivation | | | | | | |
| HM1 | 0.782 ** | 0.024 | 32.317 | <0.000 | 0.743 | 0.822 |
| HM2 | 0.793 ** | 0.023 | 33.880 | <0.000 | 0.755 | 0.832 |
| HM3 | 0.744 ** | 0.026 | 28.277 | <0.000 | 0.700 | 0.787 |
| Cue to Action | | | | | | |
| CA1 | 0.596 ** | 0.052 | 11.476 | <0.000 | 0.511 | 0.682 |
| CA2 | 0.359 ** | 0.054 | 6.640 | <0.000 | 0.270 | 0.448 |
| CA3 | 0.506 ** | 0.051 | 9.909 | <0.000 | 0.422 | 0.590 |
| Perceived Susceptibility | | | | | | |
| PSU1 | 0.535 ** | 0.107 | 5.014 | <0.000 | 0.360 | 0.711 |
| PSU2 | 0.241 ** | 0.064 | 3.763 | <0.000 | 0.135 | 0.346 |
| Perceived Severity | | | | | | |
| PSE1 | 0.595 ** | 0.042 | 14.190 | <0.000 | 0.526 | 0.664 |
| PSE2 | 0.794 ** | 0.042 | 18.956 | <0.000 | 0.725 | 0.863 |
| Perceived Barriers | | | | | | |
| PBA1 | 0.646 ** | 0.038 | 17.124 | <0.000 | 0.584 | 0.708 |
| PBA2 | 0.708 ** | 0.037 | 19.289 | <0.000 | 0.648 | 0.768 |
| Structural Model | | | | | | |
| PB ➜ Using CRS | 0.203 | 0.373 | 0.544 | 0.587 | −0.411 | 0.816 |
| HM ➜ Using CRS | −0.197 | 0.676 | −0.291 | 0.771 | −1.308 | 0.915 |
| CA ➜ Using CRS | −0.154 | 0.227 | −0.677 | 0.498 | −0.527 | 0.220 |
| PSU ➜ Using CRS | −0.519 | 2.148 | −0.242 | 0.809 | −4.053 | 3.014 |
| PSE ➜ Using CRS | 0.731 | 1.521 | 0.481 | 0.631 | −1.770 | 3.232 |
| PBA ➜ Using CRS | 1.017 | 3.396 | 0.299 | 0.765 | −4.570 | 6.603 |

Note: ** *p*-value < 0.05.

**Table 7.** Structural equation modeling results for the CRS user.

| Variable | CRS User | | | | | |
|---|---|---|---|---|---|---|
| | Estimate | S.D. | t-Stat | *p*-Value | Lower 5% | Upper 5% |
| Measurement Model | | | | | | |
| Perceived Benefits | | | | | | |
| PB1 | 0.562 ** | 0.073 | 7.665 | <0.000 | 0.441 | 0.683 |
| PB2 | 0.368 ** | 0.068 | 5.395 | <0.000 | 0.256 | 0.480 |
| PB3 | 0.413 ** | 0.068 | 6.042 | <0.000 | 0.301 | 0.526 |
| Health Motivation | | | | | | |
| HM1 | 0.741 ** | 0.031 | 24.029 | <0.000 | 0.690 | 0.792 |
| HM2 | 0.760 ** | 0.030 | 25.719 | <0.000 | 0.711 | 0.808 |
| HM3 | 0.727 ** | 0.031 | 23.353 | <0.000 | 0.676 | 0.778 |
| Cue to Action | | | | | | |
| CA1 | 0.567 ** | 0.050 | 11.264 | <0.000 | 0.484 | 0.650 |
| CA2 | 0.386 ** | 0.058 | 6.696 | <0.000 | 0.291 | 0.481 |
| CA3 | 0.662 ** | 0.050 | 13.209 | <0.000 | 0.579 | 0.744 |
| Perceived Susceptibility | | | | | | |
| PSU1 | 0.617 ** | 0.081 | 7.580 | <0.000 | 0.483 | 0.751 |
| PSU2 | 0.348 ** | 0.063 | 5.562 | <0.000 | 0.245 | 0.451 |

**Table 7.** *Cont.*

| Variable | CRS User | | | | | |
|---|---|---|---|---|---|---|
| | **Estimate** | **S.D.** | **t-Stat** | ***p*-Value** | **Lower 5%** | **Upper 5%** |
| Perceived Severity | | | | | | |
| PSE1 | 0.599 ** | 0.045 | 13.204 | <0.000 | 0.525 | 0.674 |
| PSE2 | 0.743 ** | 0.044 | 16.747 | <0.000 | 0.670 | 0.816 |
| Perceived Barriers | | | | | | |
| PBA1 | 0.616 ** | 0.044 | 14.136 | <0.000 | 0.545 | 0.688 |
| PBA2 | 0.630 ** | 0.043 | 14.526 | <0.000 | 0.558 | 0.701 |
| Structural Model | | | | | | |
| PB ➜ Using CRS | −0.247 | 0.226 | −1.096 | 0.273 | −0.619 | 0.124 |
| HM ➜ Using CRS | 0.253 | 0.416 | 0.608 | 0.543 | −0.431 | 0.937 |
| CA ➜ Using CRS | 0.404 ** | 0.184 | 2.196 | 0.028 | 0.101 | 0.707 |
| PSU ➜ Using CRS | −0.031 | 0.230 | −0.137 | 0.891 | −0.410 | 0.347 |
| PSE ➜ Using CRS | 0.607 ** | 0.249 | 2.438 | 0.015 | 0.198 | 1.017 |
| PBA ➜ Using CRS | 0.575 | 0.517 | 1.113 | 0.266 | −0.275 | 1.425 |

Note: ** *p*-value < 0.05.

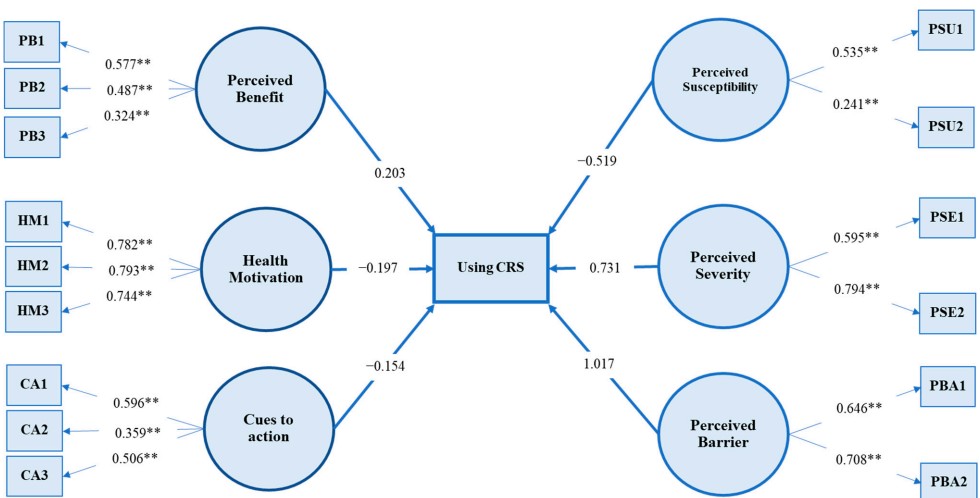

**Figure 1.** SEM of CRS non-user. Note: ** *p*-value < 0.05.

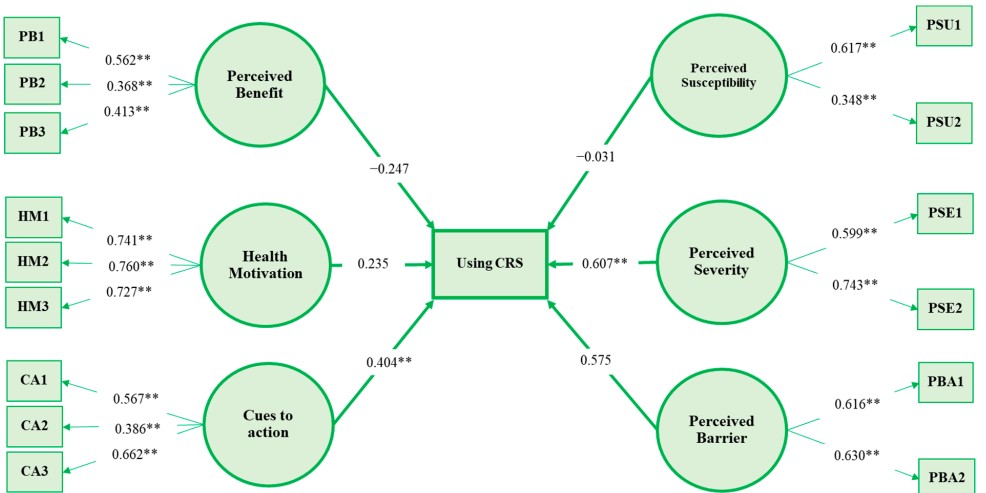

**Figure 2.** SEM of CRS user. Note: ** *p*-value < 0.05.

## 5. Discussion

This section discusses the measurement model and the structural model. The measurement model subsection explains the significant difference between CRS users and nonusers and provides preliminary recommendations. The structural model subsection discusses and elucidates the results of the latent construct correlation that affects CRS use while simultaneously comparing the two population groups.

### 5.1. Measurement Models

#### 5.1.1. Perceived Benefits (PB)

The two groups had a slight difference in perceptions (Figure 3). The CRS non-user parent group had high coefficients for PB1 and PB2. Conversely, the CRS user parent group gave the most importance to comfort following PB3. This result is quite reasonable, as parents who use CRSs tend to have various perceived benefits (such as reducing child movement and injury severity), whereas the CRS non-user group focuses on the comfort of parents when traveling. This result is consistent with several studies, such as that by Champahom et al. [10], Jomnonkwao et al. [12], and Ross et al. [67], finding that perceived benefits had a positive effect on motivation and behavior during decision-making regarding the use of safety equipment to prevent serious injuries from road crashes. In both models, PB was found to have no significant effect on CRS use.

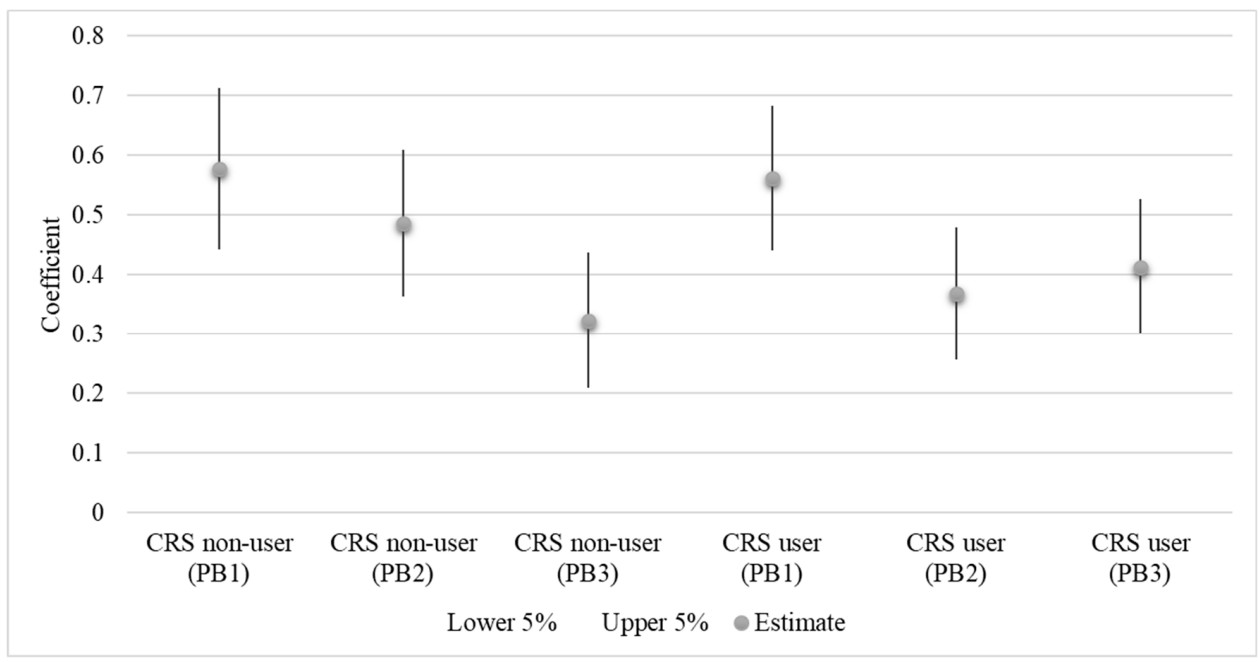

**Figure 3.** Estimated coefficients with confident intervals for perceived benefit (PB).

#### 5.1.2. Health Motivation (HM)

The two groups generated the same results. The most important variable was HM2 (coef. = 0.793 and = 0.760 for the CRS non-user and user groups, respectively), followed by HM1 and HM3. The results suggest that both parent groups care greatly about the health of their children. In addition, the attitude of giving importance to driving can increase the likelihood of using a CRS. Jomnonkwao et al. [12] found that the motivation for helmet use in urban areas was mostly due to health motivation. HM was revealed to be insignificant to CRS use in both models.

#### 5.1.3. Cues of Action (CA)

The results show a slight difference between groups (Figure 4). The highest CRS non-user variable was CA1; however, the highest CRS user variable was CA3. This clearly

reflects those CRS non-users who realize that easier accessibility is likely to increase the use of a CRS. However, the CRS users still thought that a CRS is expensive. In such a case, if the government helps subsidize CRS costs, the use of CRSs will likely increase [14]. The findings also suggest that hospitals provide CRSs to parents after their babies are born [15].

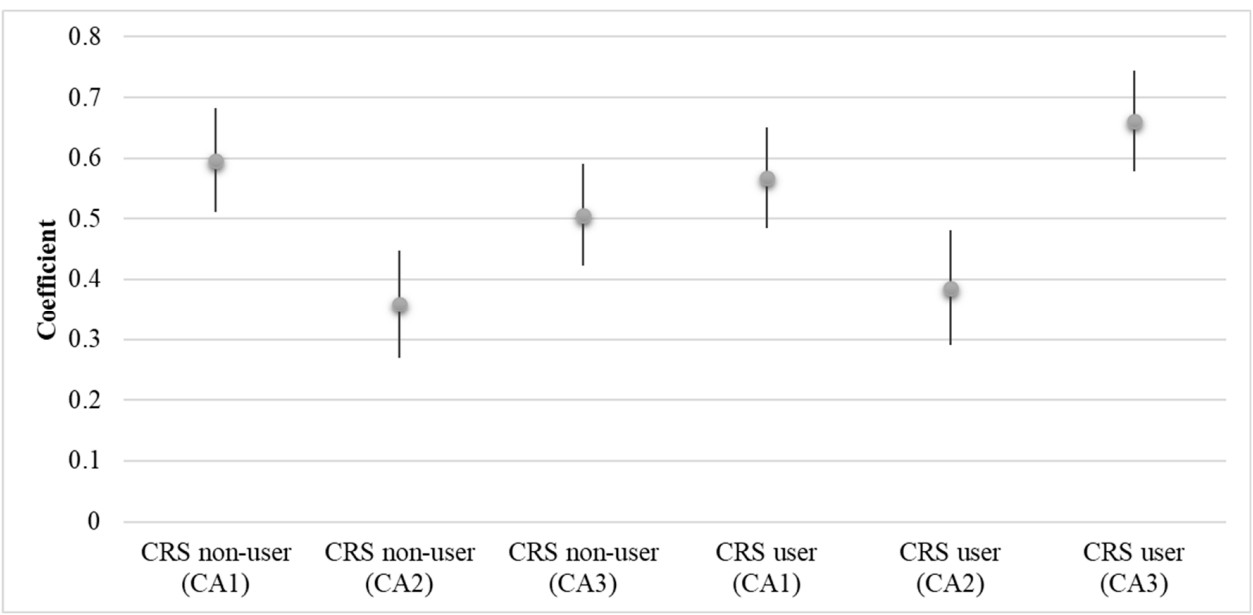

**Figure 4.** Estimated coefficients with confident intervals for cues of action (CA).

### 5.1.4. Perceived Susceptibility (PSU)

Both groups yielded similar findings. The highest coefficient variable was PSU1, while PSU2 had smaller coefficients. These observed variables are likely negative for the use of CRSs. The studies conducted by Kakefuda et al. [24] and Simpson et al. [68] stated that parents who have high perceived susceptibility tend to believe that using child safety seats is unnecessary. However, based on the results of the structural model, this factor was insignificant.

### 5.1.5. Perceived Severity (PSE)

Both groups achieved similar findings. PSE2 had high coefficients following PSE1. This result is reasonable, as many studies have supported that the use of safety seats potentially reduces accident severity. When they are aware of this issue, parents decide to use a CRS [11,12,69]. From these findings, it is suggested that by increasing the understanding of the safety benefit of CRSs and promoting that benefit, the use of CRSs could make surrounding people feel better/safer and could benefit the growth of their children.

### 5.1.6. Perceived Barriers (PBA)

Both groups obtained similar findings. PBA2 had high coefficients following PBA1. The CRS non-user parent group had more perceived barriers than the CRS user parent group. They perceived barriers with the installation and price of a CRS. The results for this group indicate that the parents perceived barriers even if they had already used a CRS. However, the barriers were still fewer than those of the CRS non-user group [22]. The results for both models showed that PBA was not significant to CRS use.

### 5.2. Structural Models

According to the results of the structural model, significant differences were observed between the two groups. In the CRS non-user group model, no factors were found to significantly affect behavior toward the use of safety seats; that is, attitudes in all six constructs were not correlated with behaviors toward CRS use. From a logical standpoint,

this may be because the increase in perceived usefulness of the CRS or accident severity possibly had no effect on the CRS-using behavior of the non-user group. In addition, this group decided not to use the CRS because they thought it was too expensive and unnecessary, among other reasons. Statistically (see Table 3), non-users' perceived severity or perceived usefulness of the CRS was not correlated in any way with the current behaviors of using the CRS (the CRS non-user group answered "never" to the question, "Do you currently use a child safety seat?" which was used for the segmentation of CRS non-users and CRS users) [70].

CRS non-users were then asked the question, "Why do you expect not to use the CRS?" using a rating scale from 1 (least agreeable) to 5 (most agreeable). The questionnaire consisted of eight question items as shown in Figure 5. The results showed that Q8 ("There is no law enforcement on child restraints") had the highest average (mean = 4.21), which is reasonable, since law enforcement will likely encourage or compel parents to use the CRS. This was also confirmed by Tessier [71]. The question with the second highest mean (4.09) was Q5 ("Poor economic conditions make it not conducive to purchasing a CRS"). This finding is also logical, because developing countries still consider the CRS expensive, which is consistent with a study by Kakefuda et al. [24], who found that expectant mothers are more likely to use the CRS if its price is reasonable.

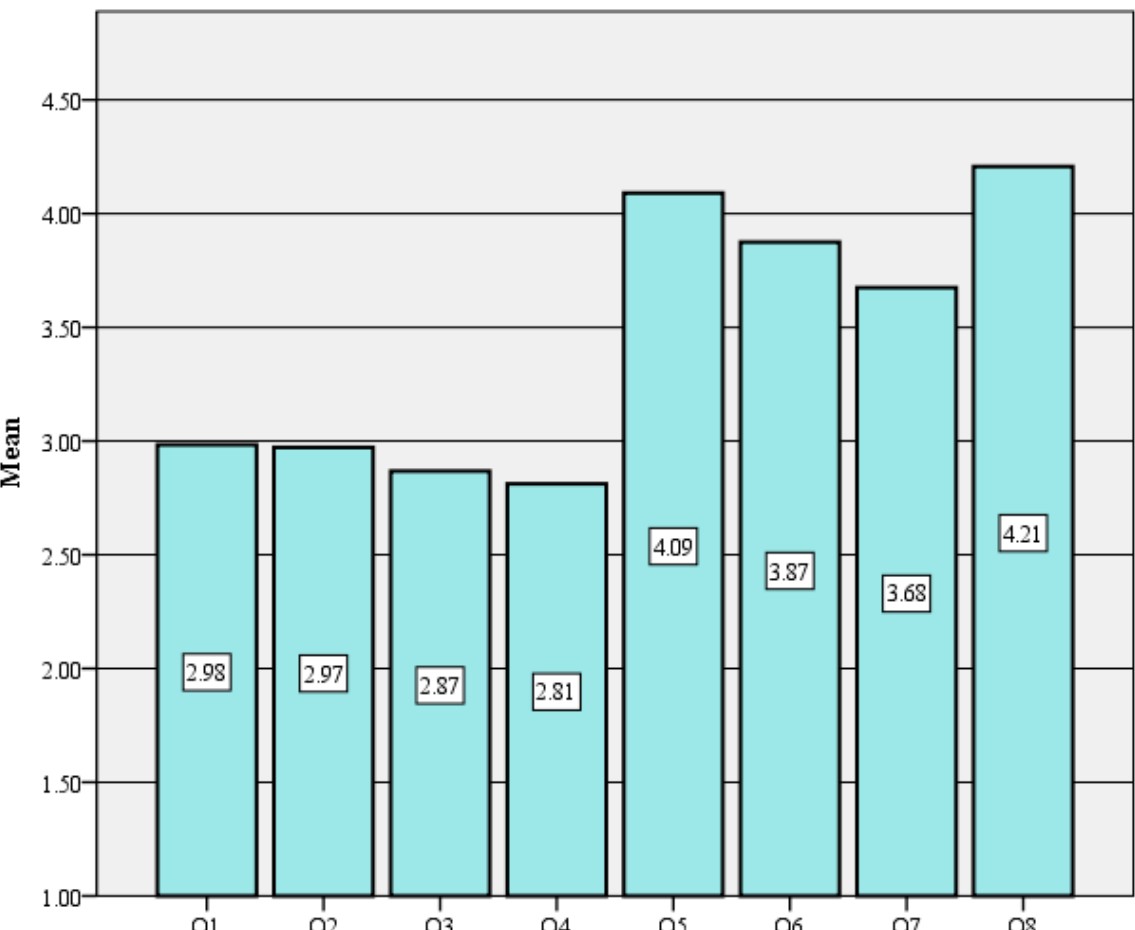

**Figure 5.** Means of rationale for not using a CRS of the parent CRS non-user. Note: Q1: Time consuming in installing; CRS is a hassle. Q2: CRS is unnecessary. Q3: Having a child sit in the back seat with seatbelt fastened is the safest way for children. Q4: I do not believe CRS will keep a child safe at the time of an accident. Q5: Poor economic conditions make it not conducive to purchasing a CRS. Q6: It is inconvenient to remove a CRS in/out every time when using it. Q7: A child is always carried in every journey. Q8: There is no law enforcement on using a child restraint system.

Meanwhile, results for the CRS user group model showed four insignificant latent variables for CRS use behavior: PB (*p*-value = 0.273), health motivation (HM) (*p*-value = 0.543), perceived susceptibility (PSU) (*p*-value = 0.891), and perceived barriers (PBA) (*p*-value = 0.266).

For the latent variable PB, the indicator PB1 had the highest loading ("I think, if using a CRS, it is not necessary to carry the child while in the car.") (coef. = 0.562). This variable was not found to significantly influence the frequency of CRS use, which is similar to the findings of Nelson et al. [72], who reported that being hands-free from a child when using a CRS did not significantly affect CRS use.

In addition, PB3 ("I think using a CRS can reduce crash injury severity.") had the second highest loading (coef. = 0.413). Studies have shown that this positive attitude should be significant to CRS use behavior [28]. However, this indicator's insignificance may be attributed to the possibility that CRS user parents have enough confidence in their own driving skills or that urban user parents can only operate their vehicles at lower speeds, thus making them think that their children would not be (seriously) injured in the case of an accident. This finding was also supported and reported by Tan et al. [15].

This study showed that HM indicators emphasize safety when driving, particularly with a child present. Many scholars have found a significant relationship between variables associated with health intentions and less risky driving behavior or the use of a seat belt [11,73]. However, as in other CRS-related studies, the current research also found no significant relationship between HM and CRS use behaviors. This may be because it is highly possible, particularly in developing countries, that parents would ask other adult occupants to hold their child or instruct their child to wear a seat belt if they were concerned about the child's safety while they were driving.

PSU contains a series of items pertaining to the reasons why parents do not consider using a CRS seat, such as driving to nearby places, the ability to avoid crashes as a result of years of driving experience, and being unimportant for experienced drivers. However, these indicators were also not found to be significant in the model, which may be because the majority of the population only drive in the city and only expect property damage or minor injuries in the case of a crash. Therefore, they are likely to think that having other adult passengers hold their child is sufficient. The provided reasons were also supported by Mora et al. [22], who found that 86% do not use a CRS because they only travel short distances.

Meanwhile, PBA is composed of a set of items regarding barriers to the use of CRSs (including cost/affordability and installation difficulties). These items were also insignificant in the CRS user model, which may be attributed to the possibility that once users own a CRS (i.e., one-time installation and not removing it unless it is necessary), it is not a barrier anymore. Similarly, Ang et al. [74] also reported that the difficulty of installing a CRS is not a problem for CRS user parents.

The CRS user model produced two statistically significant latent variables: Perceived severity (PSE) (coefficients (coef.) = 0.607, *p* = 0.015) and CA (coef. = 0.404, *p* = 0.028). PSE is measured using two indicators. The first is PSE1 ("In case of a crash where a child/children is/are not in a child restraint system, it may affect their education"), with a coefficient of 0.599, which is logical because an accident may cause severe injury or disability to a child and have long-term effects on their education and daily life, both emotionally and physically. This finding indicates that increasing parents' awareness of these issues will also likely increase their willingness to use a CRS [75].

The second significant item is PSE2 ("In case of a crash where a child/children is/are not in a safety seat, it will affect the feelings of people I know, such as parents, elder relatives, etc.") with a coefficient of 0.743. Simply put, parents who are aware of the serious emotional toll that an accidental death of a child may take on older relatives (i.e., grandparents) are more likely to adopt a CRS. This may be attributed to the fact that grandparents have the great prospect of looking forward to witnessing their grandchildren's lives and growth [24].

Therefore, increasing such awareness among parents is also likely to increase the frequency of their CRS use.

CA contains a set of positive question items that were also found to be significantly associated with the frequency of CRS use. The highest loading indicator was CA3 ("I think that the government should promote the use of child restraint systems by supporting the purchases") with a coefficient of 0.662. This finding makes sense, since CRS prices are relatively high (particularly for lower- and middle-income developing countries); therefore, parents may be more likely to purchase one if there is a price promotion. Similarly, Mora et al. [22] also reported that low-cost restraint seats will likely increase the appeal of CRS use.

CA1 ("I think that the hospitals should provide child car seats for sale/rent/loan to the mother after giving birth") also resulted in a significant parameter with a coefficient of 0.567, indicating that the frequency of CRS use is likely to increase if hospitals offer a sale, rent, or lending service to parents. This finding is consistent with the Nelson et al. [72] suggestion that hospitals must provide discounts if an infant has a CRS and that hospitals should have a CRS for rent or purchase.

Lastly, CA2 ("My close friend thinks I should use a child restraint system when I travel") was also a significant parameter, with a coefficient of 0.386. This finding indicates that if the parents' close friends have positive attitudes toward CRS use, the parents are also likely to use a CRS. This may be attributed to the effect of social norms, which are highly associated with CRS use [76].

## 6. Conclusions and Implementation

This study focused on parents' attitudes toward the use of CRSs by developing a model based on the HBM theory. The results from the EFA showed the grouped constructs. The CFA confirmed that parental attitudes consisted of six constructs. SEM was used to determine the relationship between CRS use and six constructs and one dependent variable (frequency of CRS use). The SEM results for CRS non-users indicated that there is no significant construct, whereas for CRS users it found that perceived severity and cue to action are positively significant. The researchers conducted a comparison test between the user and non-user groups. The measurement invariance principle was used to determine whether the researchers should analyze the two parent groups separately or not. The results indicated that the two parent groups should be separately analyzed. In other words, the attitudes of the two groups were statistically different.

With regard to practical implications, the current study uses the results of the significant latent factors (i.e., PSE and CA for the CRS user group model) to provide some policy recommendations. From the most significant latent variable of PSE (PSE2 "In case of a crash where a child/children is/are not in a safety seat, it will affect the feelings of people I know, such as parents, elder relatives, etc."), an increase in parents' awareness should be emphasized regarding the potential emotional consequences for grandparents (who also take care of their grandchildren) through a development-oriented program. Agencies involved in driving license training (e.g., the Department of Land Transport or private agencies that teach driving courses) should include this issue in training their subjects to potentially stimulate the awareness of serious consequences on surrounding people if something goes wrong with their children. This will likely result in an increase in CRS utilization behavior. For the variable reflecting PSE1, "In case of a crash where a child/children is/are not in a child restraint system, it may affect their education", this study suggests integrating a parents' education campaign by highlighting the topic about the severe impact on a child's future education and life; therefore, using a CRS will likely help protect children [64]. Kindergarten and primary-level schools should create billboards or posters to promote public relations or provide suggestions during parents' meetings. Central government agencies for education, such as the Ministry of Education, should provide parent invitation campaigns in affiliated schools to increase their knowledge on the

practical use of CRSs. Moreover, interventions that use a variety of media such as videos that evoke emotions and crash-test footage may also increase CRS use [77].

Regarding the last significant latent factor (CA), the most important variable is CA3, "I think that the government should promote the use of child restraint systems by supporting the purchases", which is a relatively important issue for lower- and middle-income developing countries. Because of the current cost of living and children's education, which are obligatory for parents, allocating their remaining financial resources to purchase a CRS is generally a difficult task. Based on the CA3 finding, a policy recommendation would be the use of government mechanisms. For example, state agencies such as the Ministry of Transport or the Ministry of Finance should consider subsidizing CRS prices by considering a reduction in import duties or organizing an exhibition for discounted CRSs at certain times of the year. In Malaysia, a discount of RM100 (21.65 USD) used to be offered to low-income households so that they could use an infant carrier car seat [78]. According to the effect of CA1 ("I think that hospitals should provide child car seats for sale/rent/loan to the mother after giving birth") on CRS use behavior, this study revealed that if a CRS is rented or sold at a hospital, the likelihood that parents would adopt the CRS will likely increase as well. Hospital mechanisms are potentially effective in helping to promote better use of a CRS for newborns, especially for first-time parents (Ang et al. [74]). A shop or catalog inside the hospital may provide purchase motivation to encourage parents' purchases with suitable prices compared to the market. Lastly, based on the CA2 finding ("My close friend thinks I should use a child restraint system when I travel"), the current study suggests running a campaign to stimulate CRS use targeting groups of common people such as those in the same workplace. Agencies associated with health promotion programs are recommended to manage this through media advertisements in such a way that using CRS could become the norm in today's society [76].

Meanwhile, as indicated in Figure 5, one of the main reasons that parents do not use a CRS is that there is no law that requires it (as in many developing countries). In developed countries, however, CRS laws have been effectively implemented [79]. This study was conducted in Thailand, where most people who live outside the city normally use pickup trucks for travel. These vehicles often do not have ISOFIX for a CRS installation system, and sometimes do not even have a seat belt in the rear seat (particularly in older cars). Therefore, CRS legislation should be considered to persuade parents to use a CRS. Additionally, health promotion agencies such as the Ministry of Public Health or the Ministry of the Interior (local government agencies for road safety) should enhance and promote the installation of safety seats (either ISOFIX or seat belts) through community visit programs and public relations via social media [80]. These simultaneous operations and implications can act as high-visibility enforcement campaigns [81]. The combination of community-wide information and enhanced enforcement campaigns is likely to be effective in increasing CRS use [82].

In terms of making recommendations geographically, since the current study's data collection was based on the context of a southeast Asian middle-income developing country (Thailand), countries with similar characteristics (e.g., no current law on CRSs, similar economic scale, from Asia region, etc.) may also adopt the recommendations from this study's findings in order to increase the frequency of CRS use.

## 7. Limitations and Further Research

As with any other research undertaking, this study is not without limitations. For instance, it segmented data into user and non-user groups, affecting the sample size of the model. This may raise the issue of low reliability and accuracy of the model result. Hence, some caution should be exercised in interpreting the findings. Utilizing a larger sample size could provide more precise probability estimates and would be more desired for future research.

In terms of future research directions, it would be worth the effort to extend the topic by examining the endogenous variables of motivation and decision-making behav-

iors [12]; incorporating preference heterogeneity (which may significantly vary across sample populations) to study CSR price determination and subsidization policies [14,83,84]; investigating differences in attitudes among urban, suburban, and rural parents [10]; or exploring the willingness to pay for reducing children's risk of death and serious injuries due to road crashes.

**Author Contributions:** Conceptualization, T.C. and S.J.; methodology, T.C.; software, T.C.; validation, P.R., W.T. and P.J.; formal analysis, T.C.; investigation, T.C.; resources, V.R.; data curation, T.C.; writing—original draft preparation, T.C.; writing—review and editing, P.J.; visualization, S.J.; supervision, V.R.; project administration, W.T.; funding acquisition, V.R. All authors have read and agreed to the published version of the manuscript.

**Funding:** This work was supported by (i) Suranaree University of Technology (SUT), (ii) Thailand Science Research and Innovation (TSRI), and (iii) National Science Research and Innovation Fund (NSRF) [Project: 2495400].

**Institutional Review Board Statement:** The study was conducted according to the guidelines of the Declaration of Helsinki and approved by the Ethics Committee of Suranaree University of Technology (COA No.131/2564).

**Informed Consent Statement:** Not applicable.

**Data Availability Statement:** The data presented in this study are available on request from the corresponding author. The data are not publicly available due to privacy policy restrictions.

**Conflicts of Interest:** The authors declare no conflict of interest.

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
