# Peer review of "Investigating Parents’ Attitudes towards the Use of Child Restraint Systems by Comparing Non-Users and User Parents"

_sustainability, doi:10.3390/su15042896_

Round 1

Reviewer 1 Report

-Please update your references according to the WHO 2018 report, including child restraint use.

-Please check your results in terms of family vs. relative differences, especially in terms of Table 2.

-The manuscript contains quite an amount of redundant information in the Methods and Discussion parts. The reasons for using a certain statistical method would be enough. Even though it would be appreciated, statistical background information was given unnecessarily, considering the current paper is not a methodology paper.

-Section 4.1: Please provide differences between the two groups by conducting appropriate statistical analyses rather than interpreting the results based on mean values. 

-Section 4.2: Please provide EFA details, including eigenvalues and information on how the number of factors was decided. Also, present Table 4 with full factorial loadings to increase transparency. 

-Please include group differences based on factors before testing the SEM model.

-The discussion of the manuscript needs to be improved. Even though item-by-item discussion can be meaningful and provide valuable information, the discussion consists of too much repetition, repeated presentations of items, item names, and factor loadings which were already presented in the discussion. On the contrary, model structures, comparisons of users/non-users, and implications for non-users were not discussed enough compared to other sections of the discussion.

Reviewer 2 Report

The topic is interesting, but there are some suggestions for improvement:

·        The title of article is too long. It should be shortened.

·        Problem statement requires improvement and the gaps should be related to limitations if research about the topic with support from recent references.

·        Hypotheses of the study are not presented.

·        Measurement of constructs should be presented clearly by indicating the number items used to measure each construct and the sources of items.

·        Common method bias should presented.

·         Discussion section cannot be seen in the article. It should be added to compare the results with those of past studies.

- Discriminant validity should be presented.

- No need to have sub-title sin introduction.

·         Proofreading is required for the entire paper. There are many grammatical mistakes.

Round 2

Reviewer 1 Report

I would like to thank the authors for their letter and editing.

Author Response

Thank you very mouch, the referees’ comments are beneficial to revision for improvement of our research paper. 

Reviewer 2 Report

I can see some improvement in teh paper but the following issues should be addressed:

1. The paper still requires proofreading

2. Measurement scales of constructs should be clearly presented in the methodology section. Authors should state from where the they adapted the measurement scales of constructs.

3. The author should state which software was used for data analysis. SEM has different softwares.

4. Factor loading for many items in Figure 2 are very low. Authors should justify that. Acceptable values should exceed 0.5
